# Text-based predictions of COVID-19 diagnosis from self-reported chemosensory descriptions

Hongyang Li[1], Richard C. Gerkin[2,3], Alyssa Bakke [4], Raquel Norel [1], Guillermo Cecchi [1], Christophe Laudamiel[5], Masha Y. Niv [6], Kathrin Ohla[4,7], John E. Hayes[4], Valentina Parma[8] & Pablo Meyer [1✉]

**Abstract**

**Background** There is a prevailing view that humans' capacity to use language to characterize sensations like odors or tastes is poor, providing an unreliable source of information.
**Methods** Here, we developed a machine learning method based on Natural Language Processing (NLP) using Large Language Models (LLM) to predict COVID-19 diagnosis solely based on text descriptions of acute changes in chemosensation, i.e., smell, taste and chemesthesis, caused by the disease. The dataset of more than 1500 subjects was obtained from survey responses early in the COVID-19 pandemic, in Spring 2020.
**Results** When predicting COVID-19 diagnosis, our NLP model performs comparably (AUC ROC ~ 0.65) to models based on self-reported changes in function collected via quantitative rating scales. Further, our NLP model could attribute importance of words when performing the prediction; sentiment and descriptive words such as "smell", "taste", "sense", had strong contributions to the predictions. In addition, adjectives describing specific tastes or smells such as "salty", "sweet", "spicy", and "sour" also contributed considerably to predictions.
**Conclusions** Our results show that the description of perceptual symptoms caused by a viral infection can be used to fine-tune an LLM model to correctly predict and interpret the diagnostic status of a subject. In the future, similar models may have utility for patient verbatims from online health portals or electronic health records.

**Plain language summary**

Early in the COVID-19 pandemic, people who were infected with SARS-CoV-2 reported changes in smell and taste. To better study these symptoms of SARS-CoV-2 infections and potentially use them to identify infected patients, a survey was undertaken in various countries asking people about their COVID-19 symptoms. One part of the questionnaire asked people to describe the changes in smell and taste they were experiencing. We developed a computational program that could use these responses to correctly distinguish people that had tested positive for SARS-CoV-2 infection from people without SARS-CoV-2 infection. This approach could allow rapid identification of people infected with SARS-CoV-2 from descriptions of their sensory symptoms and be adapted to identify people infected with other viruses in the future.

[1] Health Care and Life Sciences, IBM T.J. Watson Research Center, Yorktown Heights, NY, USA. [2] School of Life Sciences, Arizona State University, Tempe, AZ, USA. [3] Osmo, Cambridge, MA, USA. [4] Department of Food Science, The Pennsylvania State University, University Park, PA, USA. [5] Department of Scent Engineering, DreamAir LLC, New York, NY, USA. [6] The Faculty of Agriculture, Food and Environment, The Hebrew University of Jerusalem, Rehovot, Israel. [7] Science & Research, dsm-firmenich, Satigny, Switzerland. [8] Monell Chemical Senses Center, Philadelphia, PA, USA. ✉email: pmeyerr@us.ibm.com

Recent advances in natural language processing (NLP) models have had a clear impact on the way that computers understand, interpret, and generate human language, opening up new possibilities for the use of NLP in a wide range of applications, including healthcare[1–3]. Although NLP models are not currently part of routine clinical practice, they could represent helpful cost-effective, and easy-to-use first-aid tools to support clinicians in decision-making[4], especially when the symptoms reported are unexpected. A clear example of a situation in which the inclusion of NLP models would have helped clinical practice is represented by the coronavirus disease-19 (COVID-19). Several anecdotal reports in early March 2020 suggested new loss of smell and taste as one of the core symptoms of COVID-19. Since then, several studies confirmed that altered chemosensation in all three modalities involved in flavor perception, namely olfaction, gustation, and chemesthesis[5–9], has been experienced by the majority of individuals who contracted COVID-19[10,11]. Although direct smell testing provides better estimates of the prevalence of smell loss in the population[10], the vast majority of healthcare providers did not have and still does not have tools at hand to test for loss of smell and taste or chemesthesis in a standardized way[12]. Nevertheless, they could reliably and invariantly access verbal reports of the patients' perceptual experiences. Chemosensory self-reports have long-been considered unreliable sources of information from a research and clinical perspectives[13], yet early in the pandemic, changes in smell[11] and the analysis of text-based sources was able to pre-date the spread of COVID-19. For instance, self-reports of smell/taste changes were closely associated with hospital overload and they could represent earlier markers of the spread of infection of SARS-CoV-2 than the ones implemented by government authorities[14]. Additionally, during the initial pandemic waves, negative reviews of Yankee Candles have consistently peaked, indicating that a large group of individuals may have been unaware of their smell loss, but were able to report that products did not have the expected smell[15]. Even as the acute novel phase of the pandemic subsides, the impact of COVID-19 on those who lost their sense of smell is long-lasting and ~5–10% of such patients present persistent chemosensory loss[16–18].

Early work indicated quantitative self-reports of chemosensory function based on recall[19,20] or direct test with household items[9] can be used to predict COVID-19[19], yet they are not commonly used in healthcare. Thus, we were interested in exploring an alternative method that would match data collection with the resources available across healthcare contexts, namely the straightforward collection of written comments describing changes in perception in participants' own words when asked about their smell and taste. Further, this approach might reveal characteristics of underlying perceptual changes caused by the disease that would evade a standard chemosensory test, since the relationship between COVID-19 and human language descriptions of perceptual changes has been largely underexplored.

Here, we present a case study using a machine learning approach to detect COVID-19 using text describing perceptual changes related to chemosensory changes during COVID-19. Chemosensory losses often manifest via changes in flavor perception, as flavor arises from different sensory modalities: smell, taste and chemesthesis. The technical definition of taste is much narrower than common usage, at least in English, meaning that taste and loss of smell are often conflated[21,22]. We hypothesized that linguistic features manifesting from chemosensory changes, broadly defined, would allow us to develop a novel framework to classify individuals with and without COVID-19 only using responses to a crowd-sourced survey on smell and taste abilities. We framed this task as a sentiment analysis problem, which looks at the emotion expressed in a text implementing a higher-level classifier that divides the spectrum of emotions into positive, negative, and neutral[23–25], and applied cutting-edge Natural Language Processing (NLP) models. Indeed, recent evolutions of Large Language Models (LLMs)[26] have led to modular algorithms allowing the fine-tuning adaptation of pre-trained models on custom data. Receiver Operating Characteristic (ROC) curves are widely used in both psychophysics and medical diagnostics for binary classification; here we used the Area Under Curve of the ROC (AUC-ROC) as the evaluation metric for such a binary classification problem. To further investigate the hidden relationship between human language and COVID-19 symptoms, we also used a game-theory method (SHAP) to analyze the importance of each word in determining COVID-19 status. Following this approach, we were able to differentiate COVID-19 positive and negative subjects with an AUC-ROC of 0.65 while also finding that sentiment-related and chemosensory describing words are the most important to distinguish subjects that had the disease.

## Methods

**Data collection**. In this analysis, we used a crowdsourced dataset collected between April 7th and July 2nd 2020 by the Global Consortium for Chemosensory Research (GCCR). This dataset consists of questionnaire surveys in 30+ languages, including questions related to the perception of smell, taste, and chemesthesis before, during and after COVID-19. It was approved by the IRB of Pennsylvania State University. Participants gave informed consent to participate. As authors were part of the GCCR no license was needed to access the data and a secondary IRB approval was waived. Participants were categorized as being either positive or negative, based on self-report of lab-based test results of COVID-19. In our NLP analysis, we focused on questionnaire responses in English from 1653 participants. The majority of the participants were COVID-19 positive ($N = 1432$, 86.6%), and 13.4% of the participants were COVID-19 negative ($N = 221$). The COVID-19 negative participants were further classified into option 5 class ($N = 125$) and option 6 class ($N = 96$), depending on whether or not they had respiratory symptoms. Given some participants did not provide any text descriptions about their perception ability change, we excluded them in this study and ended up with a total of 1232 participants (positive $N = 1085$, negative option 5 $N = 89$ and option 6 $N = 58$). We used the responses to four questions as the input: (1) Comment—changes in smell, (2) Comment—changes in taste, (3) Comment—changes in chemesthesis, and (4) Comment—Anything else smell, taste, flavor.

**Cross-validation experiments**. To build machine learning models and systematically evaluate the predictive performance of our method, we partitioned the 1232 non-missing participants into the training, validating and testing subsets. We performed tenfold cross-validation experiments. Specifically, in each fold, we used 90% of the data to train the model and select hyperparameters through validation. Then the remaining 10% of the data were held out as an independent testing set to evaluate performance. The average performance from tenfold cross-validation experiments was calculated to represent the overall predictive performance of our method. We designed two scenarios: (1) using option 5 class participants as negative cases against positive cases and (2) using option 6 class participants as negative cases against positive cases. The performances of these two scenarios were calculated and reported. Since the dataset is highly unbalanced with 86.6% COVID-19 positive 13.4% COVID-19 negative participants, we randomly oversampled the negative participants so that the ratio of positive and negative samples became equal (1:1) in model training. Since the dataset is highly unbalanced with 86.6% COVID-19 positive 13.4% COVID-19 negative participants, we randomly oversampled the negative participants so that the ratio of positive and negative samples became equal (1:1) in model training.

**NLP model**. We developed natural language processing models to address the text-based predictions of COVID-19. Specifically, we leveraged the state-of-the-art Bidirectional Encoder Representations from Transformers (BERT) framework[27], RoBERTa and DistilBERT[28] a light-weighted and faster version. The word tokenizer was pre-trained and used in preprocessing the text input. Then we fine-tuned the three models, BERT, RoBERTa and DistilBERT on the GCCR dataset. The deep NLP models were implemented using "transformers" (4.21.0) in Python.

**Performance evaluation**. We used the area under the receiver operating characteristic curve (AUC-ROC) to comprehensively evaluate the predictive performance of this binary classification task. Compared to other cutoff-dependent metrics (i.e., accuracy, sensitivity, and false-positive rate), AUC-ROC is independent of the cutoff selection during classification and systematically considers the performance of each cutoff value. The baseline AUC-ROC of random prediction is 0.5.

**Statistical tests**. To evaluate the statistical differences of AUC-ROC values among different models, we performed the Wilcoxon signed-rank test. Specifically, the AUC-ROC values from tenfold cross-validation experiments were calculated from both our NLP model and the baseline. Then we ran the one-sided paired Wilcoxon test using R (4.1.3).

**Statistics and reproducibility**. To evaluate the statistical differences of AUC-ROC values among different models, we performed the Wilcoxon signed-rank test. Specifically, the AUC-ROC values from tenfold cross-validation experiments were calculated from both our NLP model and the baseline. Then we ran the one-sided paired Wilcoxon test using R (4.1.3).

**SHAP analysis**. To investigate the contribution of each word in determining the COVID-19 diagnosis, we performed the SHAP analysis of the trained DistilBERT models. For each sample in the testing set during tenfold cross-validation experiments, we calculated the SHAP value of each word in the input text. Then the absolute SHAP values of all words in all the testing samples were collected and summarized. Based on the occurrence of words, we focused on the high-frequency words that occurred in at least 10% of all testing samples. Meanwhile, the pronouns (i.e., "I") and articles (i.e., "the") were excluded from the SHAP analysis owing to their neutral sentiment. We performed this analysis for both option 5 class and option 6 class models, where the results were similar. The SHAP analysis was performed using "shap" (0.39.0) in Python.

**Reporting summary**. Further information on research design is available in the Nature Portfolio Reporting Summary linked to this article.

## Results
**Overview of experimental design**. Here, we developed a machine learning method to detect COVID-19 based on text descriptions of changes in chemosensation (Fig. 1A). We leveraged a large-scale dataset from a survey organized by the GCCR with responses collected between April and July, 2020. It consists of a crowdsourced questionnaire in participants with respiratory diseases and illnesses. The surveyed symptoms include self-reported nasal congestion and changes in smell, taste, and chemesthesis perceptual ability during or before the respiratory illness. The GCCR questionnaire contains quantitative questions that require binary or scaled responses, as well as qualitative questions such as the responses to four questions we used as input for the present analysis: (1) Comment—changes in smell, (2) Comment—changes in taste, (3) Comment—changes in

chemesthesis, and (4) Comment—Anything else smell, taste, flavor (see Fig. 1B, C). Responses were segregated based on the self-reported COVID-19 status by participants.

Surprisingly, 1232 out of 1635 participants responded to the four questions, sometimes extensively describing their chemosensory symptoms. We also noted the existence of a clear overlap in the description of chemosensory symptoms between COVID-19 positive and negative cases which may derive from the fact that upper respiratory tract infections caused by common colds and flu have similar symptoms and may also compromise olfaction, albeit only temporarily[29]. Based on this data we developed an NLP deep learning model that accepts the text as input and predicts COVID-19 status. We built a binary classification NLP model to distinguish COVID-19 positive participants from COVID-19 negative participants. The ground truth is based on the outcomes of COVID-19 lab tests, where participants were classified as either COVID-19 positive (label = 1) or COVID negative (label = 0). The predictive performance was evaluated by the area under receiver operating characteristic curve (AUC-ROC), a standard metric for binary classification problems. To understand the feature importance of each word in detecting COVID-19, we performed SHAP analysis and revealed words with high contributions and frequencies.

**Machine learning identifies COVID-19 positive participants based on text analysis**. The subset of the GCCR questionnaire responses used for the analysis initially consisted of 1653 english-speaking participants. Based on responses to Question 8—"Have you been diagnosed with COVID-19?"— the COVID-19 positive Laboratory-tested group included those that responded with either option 2 ("Yes—diagnosed with viral swab") or option 3 ("Yes—diagnosed with another lab test") and the COVID-19 negative—Lab-tested group responded with option 5 ("No, I had a negative test, but I have symptoms) or responded with option 6 ("No—I do not have any symptoms"). This dataset is highly unbalanced, since the majority of participants are COVID-19 positive (N = 1432, 86.6%) and only a small fraction are COVID-19 negative (N = 221, 13.4%). The negative participants were further sub-grouped into individuals with (N = 125, 7.6%, option 5) or without (N = 96, 5.8%, option 6) obvious respiratory symptoms, further complicating the problem. The final dataset that contained responses to the 4 comments relative to chemosensory perception consisted of 1232 participants, of which N = 1085 received a positive test result and 147 received a negative test result. This latter group was again further divided in two subgroups, depending on whether they had (option 5) or they did not have (option 6) respiratory symptoms. Two classification NLP models were developed based on these two subgroups against positive participants.

For the LLM we used a distilled version of the Bidirectional Encoder Representations from Transformers (BERT) model[27], DistilBERT[28], which is lighter and faster (Fig. 2A). Indeed, BERT was trained on 3.3B words and has 110 M parameters. The DistilBERT model and word tokenizer are pre-trained, so we fine-tuned this pre-trained model using the GCCR dataset. To evaluate the predictive performance of our model, we performed a tenfold cross-validation experiment (see Methods - Cross Validation). The baseline of random prediction of an AUC-ROC analysis is 0.5 while the average AUC-ROC for distinguishing positive cases from negative cases without respiratory symptoms (option 6 class in the survey; see Supplementary Information) is 0.65, as shown in the blue boxplot in Fig. 2B, indicating that the model was able to extract information to generate a prediction above random. As expected, the average AUC-ROC for identifying negative cases with respiratory symptoms (option 5 class) is 0.62 (the orange boxplot in Fig. 2B), which is slightly lower than option 6 class with no

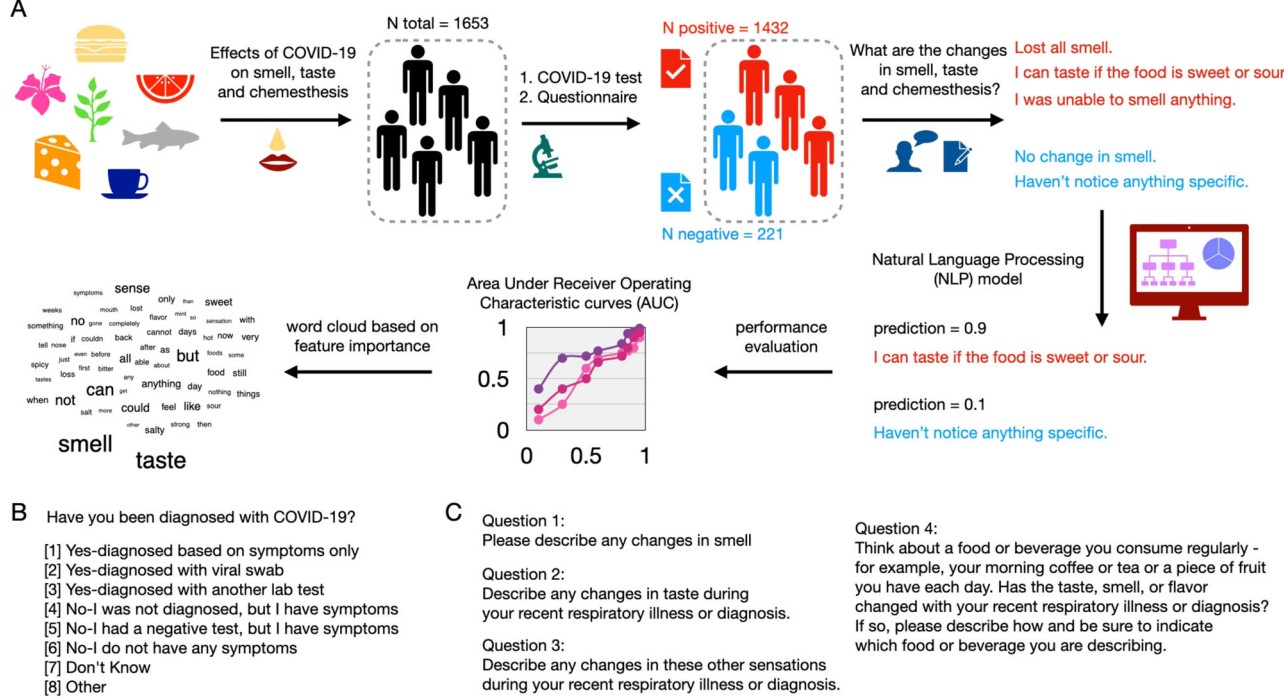

**Fig. 1 Schematic representation of predicting COVID-19 based on text. A** COVID-19 affects the perceptual abilities of humans, including smell, taste and chemesthesis. We studied a large dataset of 1653 participants in the Global Consortium for Chemosensory Research (GCCR). In the data described here, all participants reported their COVID-19 lab test result and provided text answers about changes in perceptual ability during COVID-19. We then built NLP models to analyze these text answers and distinguish COVID-19 positive from COVID-19 negative participants. The predictive performance of our method was evaluated by the Area Under Curve of the receiver operating characteristic (AUC-ROC). To dissect the importance of each word in detecting COVID-19, we further performed the SHAP analysis and highlighted the top contributing words to the classification. **B** Question relative to COVID-19 test results and symptoms. **C** The four questions relative to changes in smell, taste, and chemesthesis.

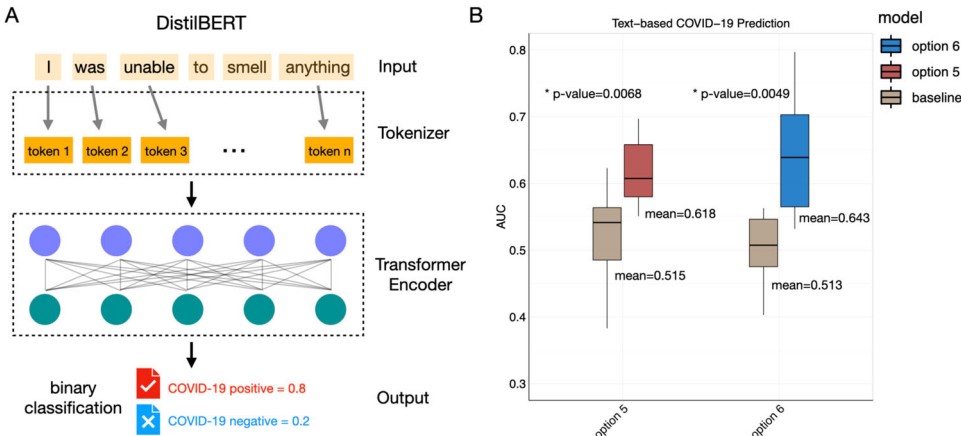

**Fig. 2 NLP model and predictive performance. A**. The DistilBERT model used for text analysis. Input text responses were first converted into tokens by the tokenizer. Then the relationship and interactions among tokens were learned by the transformer encoder. The final output is a single value between 0 and 1 in this binary classification task. **B**. The AUC-ROCs of tenfold cross-validations experiments are shown as boxplots for option 5 class and option 6 class predictions. Horizontal lines represent medians and the mean values are labeled. The whiskers represent the maximum and minimum values, whereas the bottom and top of boxes represent the first (25%) and third (75%) quartile.

respiratory symptoms. For both subjects without symptoms, option 6 class, and subjects with respiratory symptoms, option 5 class, the average and minimum AUC-ROCs from tenfold cross validations are higher than a random baseline (Supplementary Data). The random baseline AUC-ROCs are calculated by shuffling labels of testing samples as predictions, conserving the same positive ratio of the real data. This indicates free text descriptions of changes in smell, taste, and chemesthesis contain information related to

COVID-19 even when similar respiratory symptoms are present in COVID-19 positive and negative classes. In addition to DistilBERT, we also fine-tuned BERT and RoBERTa models and evaluated their performance (Supplementary Fig. 1 and Supplementary Data). For option 5 class, the three models (DistilBERT, BERT, ROBERTA) have comparable performance. For option 6 class, BERT slightly outperformed the other two models, whereas ROBERTA had a smaller variation in AUC values in tenfold cross validation when

compared to DistilBERT. Given the small differences between models and the lighter and faster usage of DistilBERT, we hereafter provide further analysis using this model.

Our NLP models and DistilBERT in particular, is indeed able to capture pertinent information to make valid predictions. For context, analysis of the same overall dataset, using a previously published logistic regression model[19] based on 0–100 visual analog scales changes in rating of smell yielded AUC-ROC of 0.72 (option 5 class) and 0.79 (option 6 class).

**Feature importance analysis reveals key words in describing perception changes in COVID-19.** To further understand how the NLP model established its predictions and reveal key words used for detecting COVID-19 positive patients, we used SHAP analysis, a popular method for explainable AI, the field of machine learning which aims to understand and explain predictions made from "black-box" models. The SHAP (SHapley Additive exPlanations) analysis is based on a game theory approach that calculates the contribution of each feature in the predictions[30]. SHAP approximates the complex non-linear decision boundary of the LLM by a linear model fit locally at the post-prediction step to understand the relative importance of the features. For each testing sample, we performed the SHAP analysis and the contributions from each word in the text descriptions were summarized. SHAP serves as a useful tool to reveal which words in the sentence led to the predicted class. We focused on the top 10% words that occur with high frequencies in the text responses. Specifically, the frequency of each word is shown as a word cloud in Fig. 3A and Supplemental Fig. 2A. In general, words related to perception as well as positive and negative sentiments have higher frequencies, including "smell", "taste", "sense", "can", "cannot", "not", "completely" and "anything". Interestingly, feature importance was lowly correlated to word frequency ($\rho < 0.11$ Supplementary Data) but these sentiment words with higher frequencies also had stronger contributions (Fig. 3B and Supplementary Fig. 2B). In addition, adjective words that describe chemosensory perceptions denoting changes in taste also considerably contribute to COVID-19 positivity predictions, such as "salty", "sweet", "spicy", and "sour". These results are consistent with previous reports showing that changes in chemosensory perception are closely related to COVID-19[19]. Interestingly, the SHAP values for the top 10% words for the two classification tasks, option 5 with respiratory symptoms and option 6 no respiratory symptoms, had a Pearson correlation ~0.7, denoting the stability of the selected features across different COVID-19 negative cohorts.

To have a numeric overview of the composition of text responses, we calculated the number of sentences per subject, the number of words per subject, and the number of words per sentence for both the COVID-19 positive (option 2/3) and negative (option 5/6) participants. The average, median, and standard deviation (SD) values are listed in Supplementary Data and the overall density distributions are shown in Supplementary Fig. 3. In general, COVID-19 negative subjects without symptoms used less sentences (5.9 sentences on average) than subjects with symptoms (6.3 sentences on average) in describing their perception changes, yet the difference is relatively small. Intriguingly, COVID-19 positive subjects also used in average 6.3 sentences just like COVID-19 negative option 5 subjects in the text response, rendering the predictive task difficult. As we expected, the average numbers of words per subject (44.7) and per sentence (10.0) are smaller for subjects without symptoms compared to the other categories. We also find that COVID-19 negative subjects with symptoms used a slightly higher number of words (58.6) than COVID-19 positive subjects (55.8).

**Case studies of words used to classify COVID-19 positive and negative participants.** To further understand—at the individual level—how machine learning models capture key words and expressions to determine COVID-19 positivity, we dissected the text responses from participants with positive (red) and negative (blue) COVID-19 test results, as shown in Fig. 4 (option 6 class) and Supplemental Fig. 4 (option 5 class). Specifically, for each participant, we calculated the absolute SHAP value from each word in the text describing perceptual changes (Supplementary Data). The SHAP value is shown as a bar plot under each word. Words with darker colors correspond to their stronger contributions in predicting COVID-19. The first two examples in Fig. 2 are COVID-19 positives and the confidence prediction scores from our model are 0.974 and 0.895 respectively (Fig. 4A, B). In this case, general comments regarding loss of taste or smell moderately indicate COVID-19 positivity.

Ideally, these text descriptions should be strong indicators or flags for COVID-19 positive results. However, COVID-19 negative participants may also have respiratory symptoms (due to rhinoviruses or other respiratory viruses), largely complicating the task. For example, two COVID-19 negative participants mentioned that "cannot taste anything" and "my lack of smell has been consistent" (Supplementary Fig. 5), rendering the classification task substantially difficult and leading to relatively high prediction scores (0.390 and 0.522) for negative cases.

Notably, we find that comments describing specific sensations have strong contributions, such as "carbonated drinks I cannot feel the tingling at all" and "I can slightly taste salt on food like

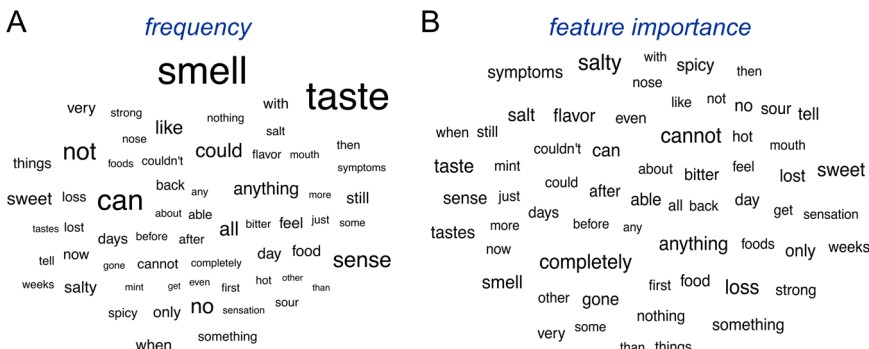

**Fig. 3 Feature importance analysis of key words in predicting COVID-19. A** The frequency of highly occurring words is shown as a word cloud for the option 6 class (no respiratory symptoms) model. The occurrence frequency is scaled to the size of the word. **B** The contributions of highly occurring words in predicting COVID-19 is shown as a word cloud for the option 6 class model. The feature importance, or absolute SHAP value, is scaled to the size of the word.

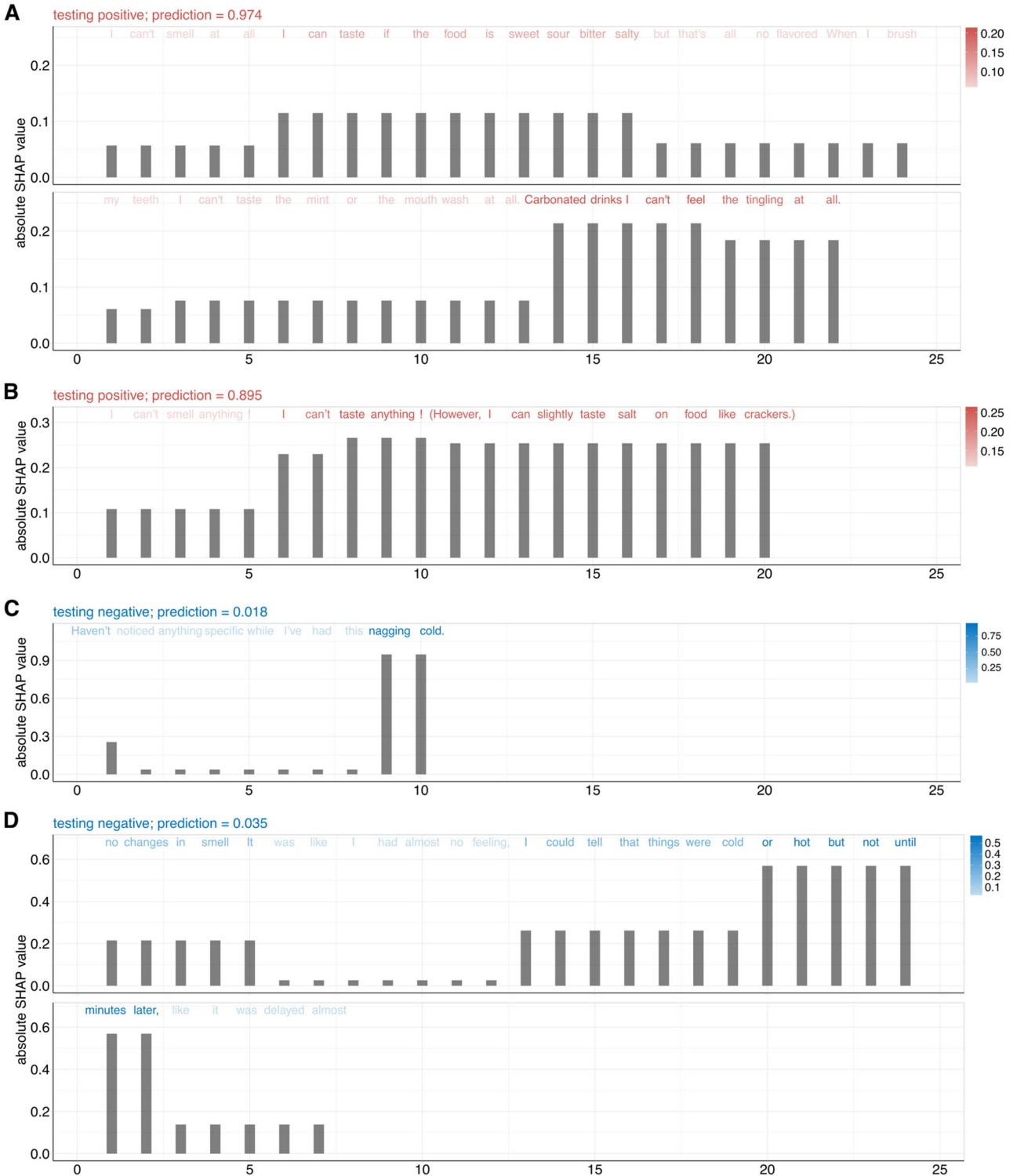

**Fig. 4 Feature importance analysis of input text responses in predicting COVID-19 positive and negative examples.** The height of the bar plot under each word as well as the color transparency correspond to the absolute SHAP value in predicting COVID-19. The SHAP values were calculated from the option 6 class model on the testing dataset. **A**, **B** The text responses of two COVID-19 positive examples are shown in red. **C**, **D** The text responses of two COVID-19 negative examples are shown in blue.

crackers" relative to chemesthesis and taste, which would not typically be affected with a classic cold virus. Finally, we analyzed two COVID-19 negative examples with prediction scores close to zero, 0.018 and 0.035. These two cases clearly mentioned that "haven't noticed anything specific" and "no changes in smell", which contributed to the predictions.

## Discussion

Here, we found DistilBERT, an LLM fine-tuned using text descriptions of changes in chemosensation, performs well at differentiating COVID-19 positive subjects from COVID-19 negative subjects without respiratory symptoms (AUC-ROC = 0.65) and subjects with respiratory symptoms but without

COVID-19 (AUC-ROC = 0.62). This is, to our knowledge, the first time that an LLM has been implemented using descriptions of changes in chemosensory perceptions to specifically predict the diagnostic status of a disease, in our case, COVID-19. Although the performance of the model is not ideal, it is thought that for complex clinical settings[31,32], such performance is sufficient to obtain clinical relevance from including chemosensory information in the diagnostic process.

Unlike other models trained using ratings obtained from long surveys, specific tasks such as smelling or tasting a specific object[9,19], our LLM classifier used free text descriptions of changes in chemosensory sensations. Importantly, although the NLP-based model underperformed the ratings model with an AUC-ROC of 0.62 versus 0.72 for the ratings (option 5 class, subjects with symptoms) and 0.65 vs 0.79 (option 6 class, subjects without symptoms), its performance was comparable. While it is now clear that COVID-19 affects chemosensory perception, the difficulty in using text-based predictions stems in part from the ambiguous descriptions of symptoms, which is rooted in the inherent openness of such responses (see Fig. 4 and Supplementary Figs. 2, 3). Ambiguous description of symptoms is probably the principal obstacle to clearly distinguish between COVID-19 positive and negative cases and may derive from the fact that upper respiratory tract infections caused by common colds and flu have similar symptoms and may also compromise olfaction, albeit only temporarily[29]. This symptom overlap has also been shown to affect chemosensory ability ratings, which led COVID-19 negative participants reporting significantly lower chemosensory abilities after then before their non-COVID illness as compared to pre-illness[8].

Apart from the simplicity of the unstructured data collection, the LLM has the advantage of allowing the analysis and selection of features that differentiate and help define characteristics of COVID-19 negative and positive subjects. Notably, the top selected features were not the most frequently used words and consisted of words such as "taste" or "smell" or sentiment-related and chemosensory describing words such as "can", "cannot", "anything", "completely" and "salty", "sweet", "spicy", "sour", "mint", "bitter".

The widespread view is that humans' capacity to characterize perceptions using language is poor and hence an unreliable source of information[33–37]. We think that our results should at least moderate this assumption as an LLM can be used to effectively classify and interpret a disease whose main symptoms are chemosensorial, using text descriptions of changes in chemosensory perceptions. This follows up previous work where we had shown that the olfactory and linguistic space have enough similarity that NLP tools can be used to predict a large set of olfactory descriptors of pure molecules[38]. We propose that similar approaches can be used to characterize and diagnose other diseases where changes in chemosensation are known to occur such as in neuropsychiatric and neurodegenerative diseases[39,40].

## Data availability
Data reported here were collected from the Global Consortium for Chemosensory Research (GCCR) core questionnaire (Appendix 1 and https://gcchemosensr.org; Parma, Ohla, et al. 2020). The numerical data underlying Figs. 2 and 4 are available in Supplementary Data.

## Code availability
The processed data code are available at: https://github.com/Hongyang449/covid19_perception/tree/main/data The code of this study is available in the GitHub repository: https://github.com/Hongyang449/covid19_perception and also in a public repository[41] https://doi.org/10.5281/zenodo.8144371.

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

## Acknowledgements

This work was supported financially with discretionary funds from the Pennsylvania State University (Penn State), including a gift from James and Helen Zallie given in support of Sensory Science at Penn State. We thank all GCCR members and the subjects that participated.

## Author contributions

H.L. performed and designed all text predictive experiments. R.G. performed the predictions in the same cohort using ratings. R.N., G.C., performed initial analysis. A.B., M.N., K.O., J.H., V.P., C.L., P.M. helped design the questionnaire and data collection. H.L., M.N., K.O., J.H., V.P., P.M. helped write and edit the manuscript. P.M. conceptualized the research and led the project.

## Competing interests
R.G. is an advisor for Climax Foods, Equity Compensation (RG); J.E.H. has consulted for for-profit food/consumer product corporations in the last 3 years on projects wholly unrelated to this study; also, he is Director of the Sensory Evaluation Center at Penn State, which routinely conducts product tests for industrial clients to facilitate experiential learning for students. P.M. is advisor of O.W. All other authors have no competing interests to declare.
