## [Peer Review File · Communications Medicine]

Reviewers' comments:

Reviewer #1 (Remarks to the Author):

First off - my apologies to the authors for the time this has taken.

Review of "Text-based predictions of COVID-19 diagnosis from self-reported chemosensory descriptions."

In this study the authors apply cutting-edge tools of Natural language processing and machine learning to distinguish online-collected text response from COVID-19 positive and COVID-19 negative test subjects. This work is part of a wide ranging effort at detecting COVID-19 and future contagions by a variety of online methods. As such it is a useful contribution to crucial public health work. The data which was used for the analysis comes from the large scale effort by the Global Consortium for Chemosensory Research (GCCR). The consortium project collected questionnaire survey in over 30 languages out of which English language data from 1232 participants fitted the requirements of this study. On the one hand this relatively large number of participants provides an opportunity for sophisticated modeling and strong results. On the other hand the data suffers from a bias of about 87% of participants being COVID-19 positive.

The classifier which the authors used is a Large language model (LLM) which is per-trained on a very large dataset of texts and then fine tuned to the text samples in this project. The performance of the classifier was assessed by reporting the area under the curve of ROC curves. The testing of the classifier was done using a 10 fold cross validation with the spread of results displayed in a box plot (figure 2). In their testing of their models the authors further divided the COVID-19 negative participants two sets of data according to the presence of respiratory symptoms. These are good choices for the evaluation of the classifier but they do raise some minor concerns which are listed below.

Minor concerns

1. As noted, 87% of participants were COVID-19 positive. Is there an attempt to control for the bias in the number of positive and negative COVID-19 subjects in the analysis ? The authors should address this concern and explain their choice or attempt a bias correcting training method.
2. The baseline results comparison in figure 2 are described as "random baseline". Please explain exactly what this random data consists of. Is it purely random data or is it shuffled data ? The correct comparison would be shuffled that which means using the data in the experiment but randomly changing the labels given to each sample.
3. There are methods of calculating p-values for ROC curves. Please use one of them to report the p-values of the results. This is a must.

Reviewer #2 (Remarks to the Author):

My expertise is in NLP and not in medical science. My comments therefore focus on the NLP part of the paper. To summarise the NLP component, you fine-tuned DistilBERT for a binary text classification task to decide on COVID and non-COVID from the responses of the patients in a questionnaire. The data used were the responses to the 4 comments relative to chemosensory perception: 1232 participants, of which N=1085 received a positive test result and 147 received a 149 negative test result. The latter negative group includes some patients with respiratory symptoms as well but not tested positive for COVID (125 in total). The fine-tuned classifier was tested using 10-fold cross-validation and the performance was reported using AUC-ROC analysis and compared against a random baseline.

I have a number of comments to this approach:

1) I would appreciate a better analysis and description of the “free text” responses of the patients. How long are these, are these sentences or key words or short phrases? What words are typically used and to what extent is this free text or triggered by the questions asked.

2) Why did you use DistilBERT and not BERT or any of the other LLMs that are available. DistilBERT is not always performing as good as BERT and BERT models have the Next Sentence Predictions as an additional learning objective. Is NSP relevant for your binary task? Would ROBERTA not be a better model? The choices here should be motivated in relation to the characteristics of the “free text” discussed in point 1)

3) I assume that random performance of the AUC-ROC analysis would be 0.5 so the system scores 0.65 and 0.62 for option class 5 (non-covid and respiratory symptoms). This is not very high. For binary NLP tasks you expect scores of at least .8 or higher. To what extent is this related to the small set of training data? Do you have evidence that more data would give results above 0.8?

4) what is the critical performance level that would make this approach useable in a clinical context? Is it 0.65 or 0.8 or higher?

5) What is the variation in performance when you applied 10-fold cross-validation? This may tell you something about the representativeness and size of the data set.

6) Why did not you implement a lexical baseline? Since English has few words for expressing chemosensory perceptions a lexical baseline could perform already very well.

7) On page 10 you state "The widespread view is that humans' capacity to characterize perceptions using language is poor and hence an unreliable source of information 31–35. We challenge this assumption by showing that an LLM can be used to interpret and classify a disease state using free text descriptions of 291 chemosensory function."  this claim needs more nuance. People can discriminate millions of smells and tastes but they do have very few words to name them. This does not mean that a LLM cannot be trained to distinguish between COVID and non-COVID using this limited vocabulary. I do not agree that your experiment falsifies our limited capacity to express chemosensory experiences.

Response to Reviewers

Referee #1: olfactory system and impact on disease, including COVID

Referee #2: NLP

Reviewers' comments:

Reviewer #1 (Remarks to the Author):

First off - my apologies to the authors for the time this has taken.

Review of “Text-based predictions of COVID-19 diagnosis from self-reported chemosensory descriptions.”

In this study the authors apply cutting-edge tools of Natural language processing and machine learning to distinguish online-collected text response from COVID-19 positive and COVID-19 negative test subjects. This work is part of a wide ranging effort at detecting COVID-19 and future contagions by a variety of online methods. As such it is a useful contribution to crucial public health work. The data which was used for the analysis comes from the large scale effort by the Global Consortium for Chemosensory Research (GCCR). The consortium project collected questionnaire survey in over 30 languages out of which English language data from 1232 participants fitted the requirements of this study. On the one hand this relatively large number of participants provides an opportunity for sophisticated modeling and strong results. On the other hand the data suffers from a bias of about 87% of participants being COVID-19 positive.

The classifier which the authors used is a Large language model (LLM) which is pre-trained on a very large dataset of texts and then fine tuned to the text samples in this project. The performance of the classifier was assessed by reporting the area under the curve of ROC curves. The testing of the classifier was done using a 10 fold cross validation with the spread of results displayed in a box plot (figure 2). In their testing of their models the authors further divided the COVID-19 negative participants two sets of data according to the presence of respiratory symptoms. These are good choices for the evaluation of the classifier but they do raise some minor concerns which are listed below.

Minor concerns

1. As noted, 87% of participants were COVID-19 positive. Is there an attempt to control for the bias in the number of positive and negative COVID-19 subjects in the analysis ? The authors should address this concern and explain their choice or attempt a bias correcting training method.

Indeed, as this reviewer notes correctly, we oversampled the COVID-19 negative participants so that the training dataset was balanced. We have added this to the revised manuscript:

“Since the dataset is highly unbalanced with 86.6% COVID-19 positive 13.4% COVID-19 negative participants, we randomly oversampled the negative participants so that the ratio of positive and negative samples became equal (1:1) in model training.”

2. The baseline results comparison in figure 2 are described as “random baseline”. Please explain exactly what this random data consists of. Is it purely random data or is it shuffled data? The correct comparison would be shuffled that which means using the data in the experiment but randomly changing the labels given to each sample.

We have added the details relative to the random baseline to the revised manuscript:
“The random baseline AUC-ROCs are calculated through randomly shuffling labels of test samples as predictions, which pertains to the same positive ratio of the real data.”

3. There are methods of calculating p-values for ROC curves. Please use one of them to report the p-values of the results. This is a must.

We added in the Methods section details about how we performed the statistical test when comparing models.

“Statistical tests

To evaluate the statistical differences of AUC-ROC values among different models, we performed the Wilcoxon signed-rank test. Specifically, the AUC-ROC values from 10-fold cross-validation experiments were calculated from both our NLP model and the baseline. Then we ran the one-sided paired Wilcoxon test using R (4.1.3).”

Reviewer #2 (Remarks to the Author):

My expertise is in NLP and not in medical science. My comments therefore focus on the NLP part of the paper. To summarise the NLP component, you fine-tuned DistilBERT for a binary text classification task to decide on COVID and non-COVID from the responses of the patients in a questionnaire. The data used were the responses to the 4 comments relative to chemosensory perception: 1232 participants, of which N=1085 received a positive test result and 147 received a 149 negative test result. The latter negative group includes some patients with respiratory symptoms as well but not tested positive for COVID (125 in total). The fine-tuned classifier was tested using 10-fold cross-validation and the performance was reported using AUC-ROC analysis and compared against a random baseline.

I have a number of comments to this approach:

1) I would appreciate a better analysis and description of the “free text” responses of the patients. How long are these, are these sentences or key words or short phrases? What words are typically used and to what extent is this free text or triggered by the questions asked.

For this revision, we ran a descriptive analysis of the text responses from three types of participants (Supplemental Table 3 and Supplemental Figure 4):

- (1) Option 6: COVID-19 negative without symptoms
- (2) Option 5: COVID-19 negative with symptoms
- (3) Option 2/3: COVID-19 positive

We first calculated the number of sentences and the number of words per participant in the text response. Meanwhile, we calculated the number of words per sentence. In general, COVID-19 negative subjects without symptoms used less sentences (5.9 sentences on average) than subjects with symptoms (6.3 sentences on average) in describing their perception changes, yet the difference is relatively small ($6.3 - 5.9 = 0.4$). Intriguingly, COVID-19 positive subjects also used the average 6.3 sentences as option 5 subjects in the text response, which rendered the

predictive task difficult. As we expected, the average numbers of words per subject (44.7) and per sentence (10.0) are smaller for subjects without symptoms compared to the other categories. We also find that COVID-19 negative subjects with symptoms used slightly more words (58.6) than COVID-19 positive subjects (55.8).

All this was added to the manuscript.

2) Why did you use DistilBERT and not BERT or any of the other LLMs that are available. DistilBERT is not always performing as good as BERT and BERT models have the Next Sentence Predictions as an additional learning objective. Is NSP relevant for your binary task? Would ROBERTA not be a better model? The choices here should be motivated in relation to the characteristics of the “free text” discussed in point 1)

Thank you for the suggestion. In this revision, we fine-tuned BERT and ROBERTA pre-trained models on the COVID-19 prediction task. The results are shown in Supplemental Figure 4. For option 5 class, the three models (DistilBERT, BERT, ROBERTA) have comparable performance. For option 6 class, BERT outperformed the other two models, whereas ROBERTA had a smaller variation in AUC values in 10-fold cross validation when compared to DistilBERT. This was added to the manuscript.

3) I assume that random performance of the AUC-ROC analysis would be 0.5 so the system scores 0.65 and 0.62 for option class 5 (non-covid and respiratory symptoms). This is not very high. For binary NLP tasks you expect scores of at least .8 or higher. To what extent is this related to the small set of training data? Do you have evidence that more data would give results above 0.8?

We do not think that more data, even if always welcome, would change the outcome, as we believe the main issue instead is the inherent quality of the data- e.g. people with common cold will also have symptoms of smell/taste loss. This makes the predictive task becomes a difficult one. We discuss the points here raised in our manuscript:

“However, COVID-19 negative participants may also have respiratory symptoms (due to rhinoviruses or other respiratory viruses), largely complicating the task. For example, two COVID-19 negative participants mentioned that “cannot taste anything” and “my lack of smell has been consistent” (Supplemental Fig. 3), rendering the classification task substantially difficult and leading to relatively high prediction scores (0.390 and 0.522) for negative cases. Notably, we find that comments describing specific sensations have strong contributions, such as “carbonated drinks I can’t feel the tingling at all” and “I can slightly taste salt on food like crackers” relative to chemesthesis and taste, which would not typically be affected with a classic cold virus.”

4) what is the critical performance level that would make this approach useable in a clinical context? Is it 0.65 or 0.8 or higher?

We thank the reviewer for making this very good point. High predictive performance >0.85 is to be expected for diagnostic tests like medical imaging, but in more complex settings, like in a complex clinical context these performances in fact turn out to be $\sim 0.6-0.7$ (see added refs 31 and 32). Indeed, expectations in such real-world environments must be reasonable and that level of performance has been shown to be sufficient to be used as a predictive tool in a clinically meaningful way. The model selected in our study, with AUC ~ 0.65 , can be regarded as a statistically accurate model which could be implemented in a clinically complex population.

We added the following paragraph:

“Indeed, although the performance of the model is not ideal, it is thought that for complex clinical settings^{31,32} such performance is sufficient to obtain clinical relevance from including chemosensory information in the diagnostic process.”

5) What is the variation in performance when you applied 10-fold cross-validation? This may tell you something about the representativeness and size of the data set.

The variation is reflected in the boxplot of 10-fold cross-validation in Figure 2B. The statistics are:

Option 6: max = 0.797; min = 0.532; mean = 0.643; median = 0.639

Option 5: max = 0.697; min = 0.551; mean = 0.619; median = 0.608

We further added to the Figure 2B legend:

“Horizontal lines represent medians and the mean values are labeled. The whiskers represent the maximum and minimum values, whereas the bottom and top of boxes represent the first (25%) and third (75%) quartile.”

Such variations are in line with our expectations for showing that the data set is representative.

6) Why did not you implement a lexical baseline? Since English has few words for expressing chemosensory perceptions a lexical baseline could perform already very well.

This work follows a previous publication (see new added ref 38 in manuscript Gutierrez et al 2019) where we had shown that the olfactory and linguistic space have enough similarity that Natural Language Processing tools can be used to predict a large set of olfactory descriptors of pure molecules without the need of extracting a lexical baseline. Moreover, we initially unsuccessfully tried to build a classifier that used as features the distance distribution between the words in the text to a set of 21 olfactory descriptors described in ref 38. We added the reference to the discussion.

7) On page 10 you state "The widespread view is that humans' capacity to characterize perceptions using language is poor and hence an unreliable source of information 31–35. We challenge this assumption by showing that an LLM can be used to interpret and classify a disease state using free text descriptions of 291 chemosensory function."  this claim needs more nuance. People can discriminate millions of smells and tastes but they do have very few words to name them. This does not mean that a LLM cannot be trained to distinguish between COVID and non-COVID using this limited vocabulary. I do not agree that your experiment falsifies our limited capacity to express chemosensory experiences.

We think that the fact that we can use LLM tools to fine-tune and effectively predict a disease whose main symptoms are chemosensory is a proof that, although we may have a limited vocabulary, the information encoded in it is sufficient for the descriptions of such symptoms to be sufficiently accurate.

We moderated our stance in the conclusion reflecting this reviewer's comment.

REVIEWERS' COMMENTS:

Reviewer #2 (Remarks to the Author):

My comments were adequately addressed in the revision and the rebuttal.

Reviewer #4 (Remarks to the Author):

We are satisfied that you have addressed out concerns.